# Text-guided Diffusion Model for 3D Molecule Generation

## Abstract

The *de novo* generation of molecules with desired properties is a critical task in fields like biology, chemistry, and drug discovery. Recent advancements in diffusion models, particularly equivariant diffusion models, have shown promise in generating 3D molecular structures. However, these models largely work under value guidance, typically conditioning on a single property value, which might limit their ability to address complex real-world requirements. To address this, we propose the text guidance instead, and introduce TEDMol, a new *Text-guided Diffusion Model for 3D Molecule Generation*. It aims to integrate the capabilities of language models with diffusion models, thereby providing a deeper level of language understanding in 3D molecule generation. Specifically, TEDMol utilizes textual conditions to guide the reverse process, enabling the adept and flexible generation of 3D molecules. Our experimental results on various tasks demonstrate that TEDMol not only enhances the stability and diversity of the generated molecules, but also excels in capturing and utilizing information derived from textual descriptions. Our approach forms a flexible and efficient text-guided molecular diffusion framework, providing a powerful tool for generating 3D molecular structures in response to complex, textual conditions. Our code is available at https://anonymous.4open.science/r/TEDMol-11E9/.

## 1 Introduction

*De novo* molecule generation aims to produce chemically viable structures with targeted properties, which is a crucial task in biology, chemistry, and drug discovery (Hajduk & Greer, 2007; Mandal et al., 2009; Pyzer-Knapp et al., 2015; Barakat et al., 2014). However, given the vast diversity of atomic species and chemical bonds, manually generating property-specific molecules is dauntingly expensive (Gaudelet et al., 2021). Addressing this, generative models for molecule generation have gained prominence recently (Alcalde et al., 2007; Anand et al., 2022; Mansimov et al., 2019; Zang & Wang, 2020; Satorras et al., 2021a; Gebauer et al., 2019). Their primary objective is exploring the molecular space to directly produce 3D molecular structures with the desired properties (Huang et al., 2023; Luo et al., 2021a; Mansimov et al., 2019).

Recent strides in diffusion models (Sohl-Dickstein et al., 2015; Ho et al., 2020), specifically equivariant diffusion models (Hoogeboom et al., 2022; Bao et al., 2023), have paved a promising path for 3D molecule generation. Essentially, they mostly introduce diffusion noise to molecular data, then learn a reverse process in either *unconditional* or *conditional* manners to denoise this corruption, thereby crafting desired 3D molecular geometries. By "unconditional", some studies (Hoogeboom et al., 2022; Huang et al., 2023) typically craft atom coordinates and types without external constraints. By "conditional", numerous efforts (Hoogeboom et al., 2022; Bao et al., 2023) operate under specific value-based conditions as shown in 1(a). For instance, EDM (Hoogeboom et al., 2022) and EEGSDE (Bao et al., 2023) take a single or a handful of specific property values (*e.g.,* polarizability $\alpha = 100$ Bohr$^3$) as conditional inputs, and generate 3D molecular conformations contingent upon these criteria.

However, we contend that value guidance specifying a single or a handful of target properties might be insufficient to capture intricate conditions. For example, searching suitable molecules in drug design usually needs multiple properties of interest (*e.g.,* simultaneously characterized by specific polarizability, orbital energy, properties like aromaticity, and distinct functional groups) (Honório

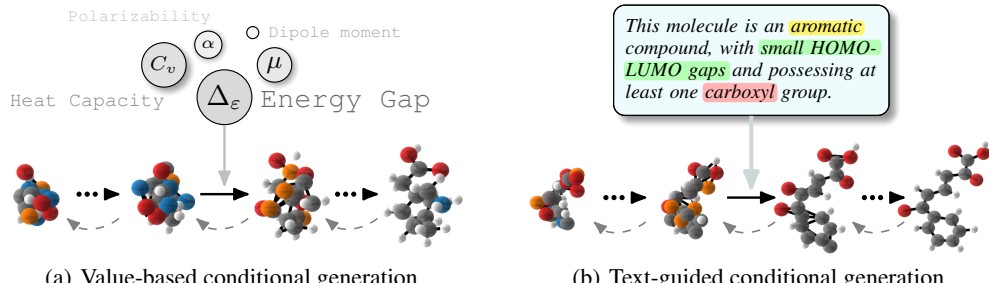

(a) Value-based conditional generation        (b) Text-guided conditional generation

Figure 1: Difference between Value- and Text-guided Diffusion Models for Molecule Generation. (a) Value guidance is typically confined to a single or a small number of properties. See Appendix B.2 for the quantum property cases. (b) Text guidance offers a flexible and generalized way to control the generation process of molecules.

et al., 2013; Gebauer et al., 2021; Lee & Min, 2022), while simple value guidance may inadequately describe such an intricate condition. In contrast, textual descriptions, such as "*This molecule is an aromatic compound, with small HOMO-LUMO gaps and possessing at least one carboxyl group.*", allow us to encompass these conditions adeptly and flexibly, even the novel properties like "aromaticity", as shown in 1(b). This motivates us to explore text guidance in diffusion models, highlighting the need for models proficient in precise language understanding and molecule generation.

Towards this end, we propose TEDMol, a new text-guided diffusion approach for 3D molecular generation. The basic idea is to combine the capabilities of the advanced language models (Devlin et al., 2019; Liu et al., 2019; Beltagy et al., 2019; Raffel et al., 2020; Brown et al., 2020; OpenAI, 2023) with high-fidelity diffusion models, enabling a sophisticated understanding of textual prompts and accurate translation into 3D molecular structures. TEDMol accomplishes this through integrating textual information with a conversion module that conditions a pre-trained equivariant diffusion model (EDM) (Hoogeboom et al., 2022), following the multi-modal fusion fashion (Su et al., 2022; Zang & Wang, 2020; Edwards et al., 2021; 2022). Specifically, at each denoising step, TEDMol first generates reference geometry, an intermediate conformation that encapsulates the textual condition signal, through a multi-modal conversion module. equipped with language and molecular encoder-decoder, corresponding to the textual condition. Then the reference geometry guides the denoising of each atom within the pre-trained unconditional EDM, gradually modifying the molecular geometry to match the condition while maintaining chemical validity. By incorporating valuable language knowledge into the high-fidelity pre-trained diffusion model, TEDMol enhances the generation of valid and stable 3D molecular conformations contingent upon a spectrum of diverse directives, without exhaustive training of the condition.

We applied TEDMol to the standard quantum chemistry dataset QM9 (Ramakrishnan et al., 2014) and a real-world text-molecule dataset from PubChem (Kim et al., 2021). The experimental results show that TEDMol accurately captures single or multiple desired properties from textual descriptions, thereby aligning the generated molecules with the desired structures. Notably, TEDMol outperforms leading diffusion-based molecule generation baselines (*e.g.,* EDM (Hoogeboom et al., 2022), EEGSDE (Bao et al., 2023)), across multiple metrics which are evident in both the stability and diversity of the generated molecules. Furthermore, when applied to real-world textual excerpts, TEDMol demonstrates its generative capability under general textual conditions. These findings suggest that TEDMol forms a flexible and efficient text-guided molecular diffusion framework, paving the way for a more profound exploration of the molecular space.

## 2 BACKGROUND

We begin with a background of diffusion-based 3D molecule generation, introducing the fundamental concepts of the diffusion model and delving into equivariant diffusion models. See the comprehensive literature review on these topics in Appendix A. In accordance with prior studies (Hoogeboom et al., 2022; Bao et al., 2023; Huang et al., 2023), we use the variable $\mathcal{G} = (\boldsymbol{x}, \boldsymbol{h})$ to represent the 3D molecular geometry. Here $\boldsymbol{x} = (x_1, \ldots, x_M) \in \mathbb{R}^{M \times 3}$ signifies the atom coor-

dinates, while $\boldsymbol{h} = (h_1, \ldots, h_M) \in \mathbb{R}^{M \times k}$ denotes the atom features. These features encompass atom types and atom charges, characterizing the atomic properties within the molecular structure.

## 2.1 DIFFUSION MODEL

The diffusion model (Sohl-Dickstein et al., 2015; Ho et al., 2020) emerges as a leading generative model, having achieved great success in various domains (Dhariwal & Nichol, 2021; Rombach et al., 2022; Ruiz et al., 2023; Song et al., 2021; Saharia et al., 2023; Schneider, 2023). Typically, it is formulated as two Markov chains: a forward process (*aka.* noising process) that gradually injects noise into the data, and a reverse process (*aka.* denoising process) that learns to recover the original data. Such a reverse process endows the diffusion model with enhanced capabilities for effective data generation and recovery.

**Forward Process.** Given the real 3D molecular geometry $\mathcal{G}_0$, the forward process yields a sequence of intermediate variables $\mathcal{G}_1, \cdots, \mathcal{G}_T$ using the transition kernel $q(\mathcal{G}_t|\mathcal{G}_{t-1})$ in alignment with a variance schedule $\beta_1, \beta_2, \ldots, \beta_T \in (0, 1)$. Formally, it is expressed as:

$$q(\mathcal{G}_t|\mathcal{G}_{t-1}) = \mathcal{N}(\mathcal{G}_t|\sqrt{1 - \beta_t}\mathcal{G}_{t-1}, \beta_t\boldsymbol{I}_n), \tag{1}$$

where $\mathcal{N}(\cdot|\cdot, \cdot)$ is a Gaussian distribution and $\boldsymbol{I}_n$ is the identity matrix. This defines the joint distribution of $\mathcal{G}_1, \cdots, \mathcal{G}_T$ conditioned on $\mathcal{G}_0$ using the chain rule of the Markov process:

$$q(\mathcal{G}_1, \cdots, \mathcal{G}_T|\mathcal{G}_0) = \prod_{t=1}^{T} q(\mathcal{G}_t|\mathcal{G}_{t-1}). \tag{2}$$

Let $\alpha_t = 1 - \beta_t$ and $\bar{\alpha}_t := \prod_{s=1}^{t} \alpha_s$. The sampling of $\mathcal{G}_t$ at time step $t$ is in a closed form:

$$q(\mathcal{G}_t|\mathcal{G}_0) = \mathcal{N}(\mathcal{G}_t|\sqrt{\bar{\alpha}_t}\mathcal{G}_0, (1 - \bar{\alpha}_t)\boldsymbol{I}_n). \tag{3}$$

Accordingly, the forward process posteriors, when conditioned on $\mathcal{G}_0$, are tractable as:

$$q(\mathcal{G}_{t-1}|\mathcal{G}_t, \mathcal{G}_0) = \mathcal{N}(\mathcal{G}_{t-1}|\widetilde{\mu}(\mathcal{G}_t, \mathcal{G}_0), \widetilde{\beta}_t\boldsymbol{I}_n), \tag{4}$$

where

$$\widetilde{\mu}(\mathcal{G}_t, \mathcal{G}_0) = \frac{\sqrt{\bar{\alpha}_{t-1}}\beta_t}{1 - \bar{\alpha}_t}\mathcal{G}_0 + \frac{\sqrt{\alpha_t}(1 - \bar{\alpha}_t)}{1 - \bar{\alpha}_t}\mathcal{G}_t, \quad \widetilde{\beta}_t = \frac{1 - \bar{\alpha}_{t-1}}{1 - \bar{\alpha}_t}\beta_t. \tag{5}$$

**Reverse Process.** To recover the original molecular geometry $\mathcal{G}_0$, the diffusion model starts by generating a standard Gaussian noise $\mathcal{G}_T \sim \mathcal{N}(\boldsymbol{O}, \boldsymbol{I}_n)$, then progressively eliminates noise through a reverse Markov chain. This is characterized by a learnable transition kernel $p_\theta(\mathcal{G}_{t-1}|\mathcal{G}_t)$ at each reverse step $t$, defined as:

$$p_\theta(\mathcal{G}_{t-1}|\mathcal{G}_t) = \mathcal{N}(\mathcal{G}_{t-1}|\mu_\theta(\mathcal{G}_t, t), \Sigma_\theta(\mathcal{G}_t, t)), \tag{6}$$

where the variance $\Sigma_\theta(\mathcal{G}_t, t) = \widetilde{\beta}_t\boldsymbol{I}_n$ and the mean $\mu_\theta(\mathcal{G}_t, t)$ is parameterized by deep neural networks with parameters $\theta$:

$$\mu_\theta(\mathcal{G}_t, t) = \widetilde{\mu}_t(\mathcal{G}_t, \frac{1}{\sqrt{\bar{\alpha}_t}}(\mathcal{G}_t - \sqrt{1 - \bar{\alpha}_t}\epsilon_\theta(\mathcal{G}_t, t))) = \frac{1}{\sqrt{\alpha_t}}(\mathcal{G}_t - \frac{1 - \alpha_t}{\sqrt{1 - \bar{\alpha}_t}}\epsilon_\theta(\mathcal{G}_t, t)), \tag{7}$$

where $\epsilon_\theta$ is a noise prediction function to approximate the noise $\epsilon$ from $\mathcal{G}_t$.

With the reverse Markov chain, we can iteratively sample from the learnable transition kernel $p_\theta(\mathcal{G}_{t-1}|\mathcal{G}_t)$ until $t = 1$ to estimate the molecular geometry $\mathcal{G}_0$.

## 2.2 EQUIVARIANT DIFFUSION MODELS

The molecular geometry $\mathcal{G} = (\boldsymbol{x}, \boldsymbol{h})$ is inherently symmetric in 3D space — that is, translating or rotating a molecule does not change its underlying structure or features. Previous studies (Thomas et al., 2018; Fuchs et al., 2020; Finzi et al., 2020) underscore the significance of leveraging these invariances in molecular representation learning for enhanced generalization. However, the transformation of these higher-order representations usually requires computationally expensive approximations or coefficients (Satorras et al., 2021b; Hoogeboom et al., 2022). In contrast, equivariant

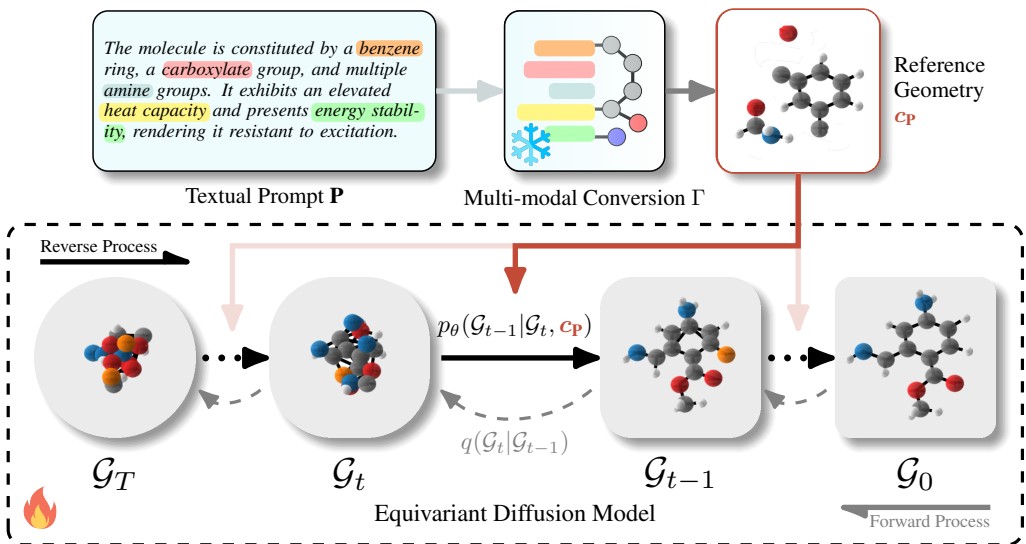

Figure 2: Framework of Our Text-guided Diffusion Model for 3D Molecule Generation (TEDMol). It iteratively generates molecules with text guidance by adopting the reference geometry $c_{\mathbf{P}}$ at each time step. The reference obtains the conditional information from the textual prompt $\mathbf{P}$ by the fixed multi-modal conversion module $\Gamma$. For the reverse process, the final molecular geometry is generated by gradually denoising the initial geometry $\mathcal{G}_T$ with the reference interfered distribution $p_\theta(\mathcal{G}_{t-1}|\mathcal{G}_t, c_{\mathbf{P}})$. Symmetrically, the forward process adds the noise with the posterior distribution $q(\mathcal{G}_t|\mathcal{G}_{t-1})$ at each step, until the molecular geometry is fully noise-corrupted when the time step is large enough. Flame 🔥 denotes tunable modules, while snowflake ❄ indicates frozen modules.

diffusion models (Köhler et al., 2020; Hoogeboom et al., 2022; Xu et al., 2022) provide a more efficient approach to ensure both rotational and translational invariance. The approach rests on the assumption that, with the model distribution $p(\mathcal{G}) = p(\boldsymbol{x}, \boldsymbol{h})$ remaining invariant to the Euclidean group E(3), identical molecules, despite being in different orientations, will correspond to the same distribution. Based on this assumption, translational invariance is achieved by predicting only the deviations in coordinate with a zero center of mass, *i.e.,* $\sum_{i=1}^{M} x_i = 0$. On the other hand, rotational invariance is accomplished by making the noise prediction network $\epsilon_\theta(\cdot)$ equivariant to orthogonal transformations (Satorras et al., 2021b; Hoogeboom et al., 2022). Specifically, given an orthogonal matrix $\boldsymbol{R}$ representing a coordinate rotation or reflection, the conformation output $a^{\boldsymbol{x}}$ from the network $\epsilon_\theta(\mathcal{G}) = \epsilon_\theta(\boldsymbol{x}, \boldsymbol{h}) = (a^{\boldsymbol{x}}, a^{\boldsymbol{h}})$ is equivariant to $\boldsymbol{R}$, if the following condition holds for all orthogonal matrices $\boldsymbol{R}$:

$$\epsilon_\theta(\boldsymbol{R}\boldsymbol{x}, \boldsymbol{h}) = (\boldsymbol{R}a^{\boldsymbol{x}}, a^{\boldsymbol{h}}). \tag{8}$$

A model exhibiting rotational and translational equivariance means a neural network $p_\theta(\mathcal{G})$ can avoid learning orientations and translations of molecules from scratch (Hoogeboom et al., 2022; Satorras et al., 2021b). In this paper, we parameterize the noise prediction network $\epsilon_\theta$ using an E(n) equivariant graph neural network as introduced by (Satorras et al., 2021b), which is a type of Graph Neural Network (Hamilton et al., 2017) that satisfies the above equivariance constraint to E(3).

## 3 TEXT-GUIDED DIFFUSION MODEL FOR 3D MOLECULE GENERATION

In this section, we elaborate on the proposed text-guided diffusion model for 3D molecule generation (TEDMol), as illustrated in Figure 2. It integrates the textual information (*i.e.,* text guidance) into the conditional signal of diffusion models by employing the reference geometry that is described in Section 3.1. Subsequently, we introduce an efficient learning approach that incorporates both the encoded conditional signal and pre-trained unconditional signal in the reverse process, to generate molecules that are not only structurally stable and chemically valid but also align well with the specified conditions, as presented in Section 3.2.

### 3.1 Integrating Textual Prompts into 3D Molecular Reference Geometry

To ensure high-fidelity 3D molecule generation, the reverse process of the diffusion model is typically guided by tailored conditional information representing desired properties like unique polarizability. We represent this conditional information as $c$, which allows us to formulate the conditional reverse process as:

$$p_\theta(\mathcal{G}_{t-1}|\mathcal{G}_t, c) = \mathcal{N}(\mathcal{G}_{t-1}|\mu_\theta(\mathcal{G}_t, c, t), \widetilde{\beta}_t \boldsymbol{I}_n) \qquad (9)$$

Unlike previous approaches relying on limited value guidance (*i.e.,* property values), in this work, we aim to steer the reverse process with text guidance (*i.e.,* informative textual descriptions), which can convey a broader range of conditional requirements. Intuitively, utilizing textual descriptions to specify conditional generation criteria not only provides greater expressivity but also better aligns the resulting 3D molecules with diverse and complex expectations.

Practically, we first introduce a textual prompt $\mathbf{P}$ describing desired 3D molecule properties. A multi-modal conversion module $\Gamma$, pre-trained on 300K text-molecule pairs from PubChem, is then employed. This module is comprised of a GIN molecular graph encoder (Xu et al., 2019; Liu et al., 2022) and a language encoder-decoder extended from BERT (Devlin et al., 2019; Zeng et al., 2022). It converts $\mathbf{P}$ into a reference geometry $\boldsymbol{c_P}$, extracting specific information from the target conditions and refining the textual condition signal:

$$\boldsymbol{c_P} = \Gamma(\mathbf{P}). \qquad (10)$$

Nevertheless, we should emphasize that valid and stable 3D molecules can hardly be obtained directly from $\boldsymbol{c_P}$. The chemical fidelity in 3D molecular space may not be guaranteed. In what follows, we describe how to utilize $\boldsymbol{c_P}$ for conditioning a pre-trained diffusion model to generate molecules that align with the desired properties, meanwhile alleviating the exhaustive training from scratch.

### 3.2 Conditioning with the Reference of Text Guidance

To leverage $\boldsymbol{c_P}$ for text-guided conditional generation while preserving the validity and stability of the synthesized molecule, TEDMol employs the iterative latent variable refinement (ILVR) (Choi et al., 2021) to condition a pre-trained unconditional diffusion model meanwhile maintaining inherent domain knowledge in the unconditional model.

With the pre-trained unconditional diffusion model EDM (Hoogeboom et al., 2022), we could perform a step-by-step reverse process. Formally, at step $t$, we can sample an unconditional proposal molecular geometry:

$$\widetilde{\mathcal{G}}_{t-1} \sim \widetilde{p}_{\widetilde{\theta}}(\widetilde{\mathcal{G}}_{t-1}|\mathcal{G}_t). \qquad (11)$$

where $\widetilde{\theta}$ is the fixed parameters of the pre-trained unconditional diffusion model (Hoogeboom et al., 2022). Then, to incorporate the condition signal $\boldsymbol{c_P}$ in the reverse process, we introduce a linear operation $\varphi_\theta(\cdot)$. Therefore the conditional denoising for one step at step $t$ can be formulated as:

$$\mathcal{G}_{t-1} = \varphi_\theta(\boldsymbol{c_P}) + (\mathcal{I} - \varphi_\theta)(\widetilde{\mathcal{G}}_{t-1}), \qquad (12)$$

where $\mathcal{I}(\cdot)$ is the identity operation and $(\mathcal{I} - \varphi_\theta)(\cdot)$ is the residual operation *w.r.t.* $\varphi_\theta(\cdot)$ (James & Wilkinson, 1971). Accordingly, the condition signal $\boldsymbol{c_P}$ is projected into the reverse denoising process by $\varphi_\theta(\cdot)$, thus $\mathcal{G}_{t-1}$ is obtained as the generated 3D molecular geometry conditioned on $\boldsymbol{c_P}$. Conceptually, the proposal geometry from unconditional generation $\widetilde{\mathcal{G}}_{t-1}$ tries to push the atoms into a chemically valid position, while the reference geometry $\boldsymbol{c_P}$ pulls the atoms towards the structure targeted to the condition.

By matching latent variables following Equation 12, we enable text-guided conditional generation with the unconditional diffusion model. Accordingly, the one-step denoising distribution conditioned on textual guidance at each step $t$ can be reformulated as:

$$\mathcal{G}_{t-1} \sim p_\theta(\mathcal{G}_{t-1}|\mathcal{G}_t, \boldsymbol{c_P}). \qquad (13)$$

Table 1: Comparison of MAE for the Generated Molecules Targeted to Desired Property. Statistics of baselines are from their original papers. The performance of EEGSDE varies depending on the scaling factor, and we report its best results. **Boldface** indicates the best performance.

| Method | MAE↓ | | | | | |
|---|---|---|---|---|---|---|
| | $C_v$ $\left(\frac{\text{cal}}{\text{mol}}\text{K}\right)$ | $\mu$ (D) | $\alpha$ (Bohr$^3$) | $\Delta\varepsilon$ (meV) | $\varepsilon_{\text{HOMO}}$ (meV) | $\varepsilon_{\text{LUMO}}$ (meV) |
| U-Bound | 6.857 | 1.616 | 9.01 | 1470 | 645 | 1457 |
| #Atoms | 1.971 | 1.053 | 3.86 | 866 | 426 | 813 |
| EDM | 1.101 | 1.111 | 2.76 | 655 | 356 | 584 |
| EEGSDE | 0.941 | **0.777** | 2.50 | 487 | 302 | 447 |
| TEDMol | **0.847** | 0.840 | **2.24** | **443** | **279** | **412** |
| L-Bound | 0.040 | 0.043 | 0.10 | 64 | 39 | 36 |

## 3.3 TRAINING OBJECTIVE

To guarantee the quality of the generated molecules, the key lies in optimizing the variational lower bound (ELBO) of negative log-likelihood, which equals minimizing the Kullback-Leibler divergence between the joint distribution of the reverse Markov chain $p_\theta(\mathcal{G}_0, \mathcal{G}_1, \cdots, \mathcal{G}_T)$ and the forward process $q(\mathcal{G}_0, \mathcal{G}_1, \cdots, \mathcal{G}_T)$:

$$\mathbb{E}\left[-\log p_\theta(\mathcal{G}_0|\boldsymbol{c}_\mathbf{P})\right] \leq -\log \sum_{t \geq 1} \underbrace{D_{\text{KL}}\left(q(\mathcal{G}_{t-1}|\mathcal{G}_t, \mathcal{G}_0)||p_\theta(\mathcal{G}_{t-1}|\mathcal{G}_t, \boldsymbol{c}_\mathbf{P})\right)}_{:=\mathcal{L}_{t-1}} + C, \qquad (14)$$

where $C$ is a constant independent of $\theta$. Note that we set $\mathcal{L}_0 = -\log p_\theta(\mathcal{G}_0|\mathcal{G}_1)$ as a discrete decoder following Ho et al. (2020). Further adopting the reparameterization from (Ho et al., 2020), $\mathcal{L}_{t-1}$ can be simplified to:

$$\mathcal{L}_{t-1} = \mathbb{E}_{\text{P}, \mathcal{G}_0, \epsilon}\left[||\epsilon - \epsilon_\theta(\sqrt{\bar{\alpha}_t}\mathcal{G}_0 + \sqrt{1-\bar{\alpha}_t}\epsilon, t, \boldsymbol{c}_\mathbf{P})||^2\right]. \qquad (15)$$

## 4 EXPERIMENTS

In this section, we present the experimental results of our proposed TEDMol model, showcasing its ability to generate molecules with desired properties. To evaluate our model, we employ the QM9 dataset (Ramakrishnan et al., 2014), which is a standard benchmark containing quantum properties and atom coordinates of over 130K molecules, each with up to 9 heavy atoms (C, N, O, F). To train our model under the condition of textual descriptions, we have curated a subset of molecules from QM9 and associated them with real-world descriptions sourced from PubChem (Kim et al., 2021). We consider PubChem as a rich source of molecular graph-language pairs, given its status as one of the most comprehensive databases for molecular descriptions. This database aggregates extensive annotations from a diverse array of sources, such as ChEBI (Degtyarenko et al., 2008), LOTUS (Rutz et al., 2022), and T3DB (Wishart et al., 2014). Each of these sources offers an emphasis on the physical, chemical, or structural attributes of molecules. Additionally, we have employed a set of textual templates to generate corresponding descriptions based on the quantum properties of the molecules, thereby enriching the content of the dataset and supplementing textual context for those molecules lacking real-world descriptions. This process has enriched QM9 into a dataset of chemical molecule-textual description pairs.

### 4.1 EXPERIMENT ON SINGLE QUANTUM PROPERTIES CONDITIONING

Following EDM (Hoogeboom et al., 2022), we first evaluate our TEDMol on the task of generating molecule conditioning on a single desired quantum property in QM9. Then we compare our TEDMol with several baselines to demonstrate the effectiveness of our model on single quantum properties conditioning molecule generation.

**Setup.** We follow the same data preprocessing and partitions as in EDM (Hoogeboom et al., 2022), which results in 100K/18K/13K molecule samples for training/validation/test respectively. In order

Table 2: Comparison of Novelty (Novel, %), Atom Stability (A. Stable,%), and Molecule Stability (M. Stable,%) on Generated Molecules Targeted to the Desired Property. Statistics of baselines are from EEGSDE. The performance of EEGSDE varies depending on the scaling factor, and we report its best results. **Boldface** indicates the best performance.

| Method | Novel↑ | A. Stable↑ | M. Stable↑ | Method | Novel↑ | A. Stable↑ | M. Stable↑ |
|---|---|---|---|---|---|---|---|
| | Condition on $C_v$ $\left(\frac{\text{cal}}{\text{mol}}\text{K}\right)$ | | | | Condition on $\mu$ (D) | | |
| EDM | 83.64 | 98.25 | 80.82 | EDM | 83.93 | 98.17 | 80.25 |
| EEGSDE | 83.78 | 98.25 | **80.83** | EEGSDE | 84.62 | 98.18 | 80.25 |
| TEDMol | **83.82** | **98.27** | **80.83** | TEDMol | **84.88** | **98.22** | **80.31** |
| | Condition on $\alpha$ (Bohr$^3$) | | | | Condition on $\Delta\varepsilon$ (meV) | | |
| EDM | 83.93 | 98.30 | 81.95 | EDM | 84.35 | 98.17 | 79.61 |
| EEGSDE | 85.17 | 98.18 | 80.99 | EEGSDE | 84.77 | **98.19** | 79.81 |
| TEDMol | **85.82** | **98.42** | **82.03** | TEDMol | **84.92** | **98.19** | **79.82** |
| | Condition on $\varepsilon_{\text{HOMO}}$ (meV) | | | | Condition on $\varepsilon_{\text{LUMO}}$ (meV) | | |
| EDM | 84.56 | 98.13 | 79.33 | EDM | 84.62 | 98.26 | 81.34 |
| EEGSDE | 84.45 | **98.26** | 80.95 | EEGSDE | 84.83 | 98.27 | 81.23 |
| TEDMol | **84.58** | 98.22 | **80.97** | TEDMol | **84.90** | **98.31** | **81.40** |

to assess the quality of the conditional generated molecules *w.r.t.* to the desired properties, we use the property classifier network $\phi_p$ introduced by Satorras et al. (2021b). Then for the impartiality, the training partition is further split into two non-overlapping halves $\mathbb{D}_a$ and $\mathbb{D}_b$ of 50K molecule samples each. The property classifier network $\phi_p$ is trained on the first half $\mathbb{D}_a$, while our TEDMol is trained on the second half $\mathbb{D}_b$. This ensures that there is no information leak and the property classifier network $\phi_p$ is not biased towards the generated molecules from TEDMol. Then $\phi_p$ is evaluated on the generated molecule samples from TEDMol as we introduce in the following.

**Metrics.** Following Hoogeboom et al. (2022), we use the mean absolute error (MAE) between the properties of generated molecules and the ground truth as a metric to evaluate how the generated molecules align with the condition (see Appendix B.1 for details). We generate 10K molecule samples for the evaluation of $\phi_p$, following the same protocol as in EDM. Additionally, we then measure novelty (Simonovsky & Komodakis, 2018), atom stability (Hoogeboom et al., 2022), and molecule stability (Hoogeboom et al., 2022) to demonstrate the fundamental molecule generation capacity of the model (also see Appendix B.1 for details).

**Baseline.** We compare our TEDMol with a direct baseline conditional EDM (Hoogeboom et al., 2022) and a recent work EEGSDE which takes energy as guidance (Bao et al., 2023). We also compare two additional baselines "U-bound" and "#Atoms" introduced by Hoogeboom et al. (2022). In the "U-bound" baseline, any relation between molecule and property is ignored, and the property classifier network $\phi_p$ is evaluated on $\mathbb{D}_b$ with shuffled property labels. In the "#Atoms" baseline, the properties are predicted solely based on the number of atoms in the molecule. Furthermore, we report the error of $\phi_p$ on $\mathbb{D}_b$ as a lower bound baseline "L-Bound".

**Results.** We generate molecules with textual descriptions targeted to each one of the six properties in QM9, which are detailed in Appendix B.2. As presented in Table 1, our TEDMol has a lower MAE than other baselines on five out of the six properties, suggesting that the molecules generated by TEDMol align more closely with the desired properties than other baselines. The result underscores the proficiency of TEDMol in exploiting textual data to guide the conditional *de novo* generation of molecules. Moreover, it highlights the superior congruence of the text-guided molecule generation via the diffusion model with the desired property, thus showing significant potential. Furthermore, as indicated in Table 2, our proposed TEDMol exhibits commendable performance in terms of novelty and stability. The text guidance we introduced has transformed the exploration of the model in the molecule generation space, generally enhancing the novelty of the generated molecules while maintaining their stability.

Table 3: Comparison of MAE, Novelty (Novel, %), Atom Stability (A. Stable,%), and Molecule Stability (M. Stable,%) on the Generated Molecules Targeted to the Multiple Desired Properties. Statistics of baselines are from EEGSDE. **Boldface** indicates the best performance.

| Method | MAE1$\downarrow$ | MAE2$\downarrow$ | Novel$\uparrow$ | A. Stable$\uparrow$ | M. Stable$\uparrow$ |
|---|---|---|---|---|---|
| Condition | $C_v$ $\left(\frac{\text{cal}}{\text{mol}}\text{K}\right)$ and $\mu$ (D) | | | | |
| EDM | 1.079 | 1.156 | 85.31 | **98.00** | **77.42** |
| EEGSDE | 0.981 | 0.912 | 85.62 | 97.67 | 74.56 |
| TEDMol | **0.645** | **0.836** | **85.79** | 97.89 | 77.33 |
| Condition | $\alpha$ (Bohr$^3$) and $\mu$ (D) | | | | |
| EDM | 2.76 | 1.158 | 85.06 | 97.96 | 75.95 |
| EEGSDE | 2.61 | 0.855 | 85.56 | 97.61 | 72.72 |
| TEDMol | **2.27** | **0.809** | **85.64** | **98.01** | **75.97** |
| Condition | $\Delta\varepsilon$ (meV) and $\mu$ (D) | | | | |
| EDM | 683 | 1.130 | 85.18 | 98.00 | 77.96 |
| EEGSDE | 563 | 0.866 | 85.36 | 97.99 | 77.77 |
| TEDMol | **489** | **0.843** | **85.44** | **98.06** | **78.03** |

## 4.2 EXPERIMENT ON MULTIPLE QUANTUM PROPERTIES CONDITIONING

The capacity to generate molecules, guided by multiple conditions, is a crucial aspect of the molecule generation model. When guided by textual descriptions, characterizing the condition with multiple desired properties is highly intuitive and flexible. Following the same setup and metrics in Section 4.1, we evaluate our TEDMol on the task of generating molecules with multiple desired quantum properties in QM9. Then we compare TEDMol with two baselines to showcase the effectiveness of our model in generating molecules conditioned on multiple quantum properties.

As shown in Table 3, our TEDMol has a remarkably lower MAE than the other two baselines, thereby demonstrating the superiority of our model in generating molecules with multiple desired properties. This also further substantiates that, without necessitating additional targeted interventions, textual conditions can be utilized in our model to guide molecule generation that conforms to multiple desired properties.

Additionally, as highlighted in Table 3, our proposed TEDMol maintains superior performance in terms of novelty and stability, when generating molecules targeted at multiple desired properties. The results indicate that the flexible integration of multiple conditions through textual description does not compromise the stability of the generated molecules. Furthermore, this approach enhances novelty when compared to the baseline.

## 4.3 GENERATION ON REAL-WORLD TEXTUAL DESCRIPTIONS

To further assess our model, we undertake additional training on a vast dataset of over 330K text-molecule pairs we gleaned from PubChem (Kim et al., 2021). Then, we generate molecules based on authentic textual excerpts from the real world to observe the capacity of our model to generate from generalized textual conditions.

Visual observations, as depicted in Figure 3, illuminate the impressive aptitude of our TEDMol in aligning molecule structures with the desired property within the textual descriptions. For instance, when the textual description includes affirmatively mentioned terms such as "polydentate macrocyclic" and "cationic affinity", the generated molecules consistently exhibit macrocyclic structures with no fewer than 10 atoms, and also possesses multiple dentate configurations, the electron cloud distribution of which is conducive to alkali metal ion binding. Moreover, when the textual description includes "polycyclic heteroarene" and specifies the solubility and heat capacity of the molecule, TEDMol generates a variety of polycyclic aromatic hydrocarbon molecules. The ubiquitously present amino and nitro groups attest to a certain degree of solubility of the molecules.

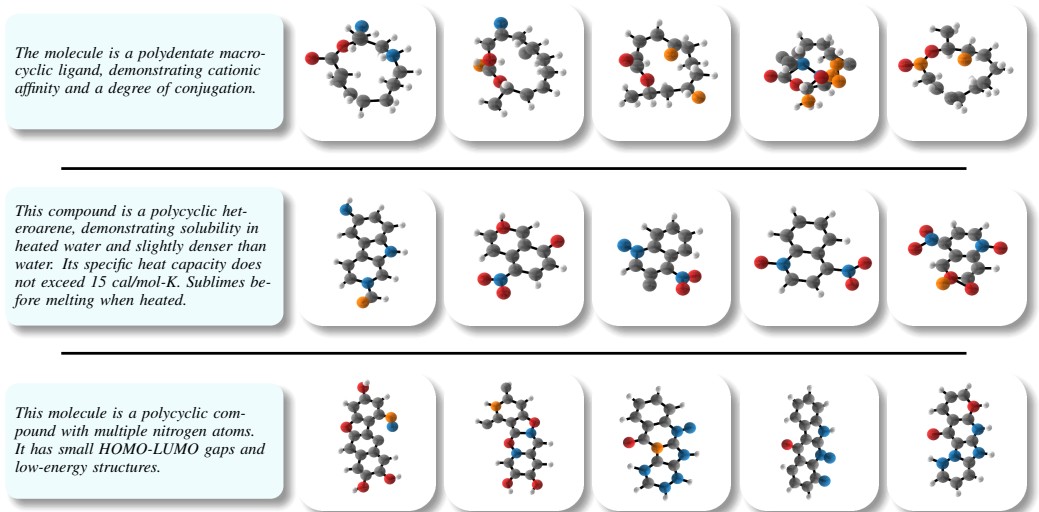

Figure 3: Generated molecules targeted to text description excerpts.

Referring to structurally similar real molecules, their expected specific heat capacity is also relatively low. Lastly, when the text description explicitly demands multiple nitrogen atoms and a low energy gap, the molecules generated by TEDMol not only possess the required polycyclic structure and multiple nitrogen atoms, but the rings on the same plane denote the low-energy structures of these molecules that are difficult to excite.

The remarkable alignment between the conditions and the generated molecule stands as a testament to the exceptional generative capabilities of TEDMol. The result demonstrates that TEDMol is equipped to deeply explore the chemical molecular space in a text-guided manner, thereby generating prospective molecules for subsequent applications. This capability could potentially expedite drug design and the discovery of materials.

## 5 CONCLUSION

In this work, we presented TEDMol, a text-guided diffusion approach for 3D molecule generation. TEDMol combines the strengths of advanced language models with high-fidelity diffusion models, thereby enabling the translation of complex textual prompts into accurate molecular structures. By integrating textual information with the denoising process of a pre-trained equivariant diffusion model, TEDMol effectively generates valid and stable molecular conformations, aligning closely with diverse textual directives. Our experiments on the QM9 and PubChem datasets demonstrated the superior performance of TEDMol over leading baselines, affirming its efficacy in capturing desired properties from textual descriptions and generating corresponding valid molecules. TEDMol presents an initial step towards a profound exploration of the molecular space, paving the way for future advancements in text-guided molecule generation.

## 6 LIMITATIONS

Despite the promising results, it is essential to acknowledge a few limitations: 1) Limited High-Quality Data. The scarcity of high-quality data comprising real-world 3D molecules with their respective textual annotations has restricted our work on fully training the model with extensive text-3D molecule pairs. 2) Sampling Efficiency. The sampling process is slow with the iteration of the total diffusion steps. This inefficiency may become a bottleneck when applying the model to large-scale applications. In our future work, we will address these challenges. Specifically, we will expand and improve the quality of available data for text-molecule pairs, which enables more comprehensive training of the model. Additionally, we will explore strategies to enhance the sampling efficiency, making the model more practical for large-scale applications.

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

# A RELATED WORKS

**Diffusion models** are initially proposed by Sohl-Dickstein et al. (2015). The basic idea is to corrupt data with diffusion noise and learn a neural diffusion model to reconstruct data from noise. Recently, they have been theoretically enhanced by establishing connections to score matching and stochastic differential equations (SDE) (Ho et al., 2020; Song et al., 2021). Such theoretical enhancements have facilitated the successful application of diffusion models across diverse domains, including image and waveform generation (Dhariwal & Nichol, 2021; Rombach et al., 2022; Chen et al., 2021; Kong et al., 2021), and have recently gained attention in the molecular sciences field (Hoogeboom et al., 2022; Huang et al., 2023; Xu et al., 2022).

**Molecule generation** is to explore the molecular space and generate novel molecules. Prior efforts (Weininger, 1988; Kotsias et al., 2020; Jin et al., 2018) often generate simplified representations of molecules, such as 1D SMILES strings and 2D molecule graphs. Some studies (Jing et al., 2022) have also tried to generate torsion angles in a given 2D molecular graph for the conformation generation task. More recently, several works (Nesterov et al., 2020; Gebauer et al., 2019; Satorras et al., 2021a; Hoffmann & Noé, 2019; Hoogeboom et al., 2022) attempt to model molecules as 3D objects via deep generative models. Diverse model architectures are proposed, including, but not limited to, variational autoencoders (Kusner et al., 2017; Dai et al., 2018; Jin et al., 2018; Simonovsky & Komodakis, 2018; Liu et al., 2018), normalizing flows (Madhawa et al., 2019; Zang & Wang, 2020; Luo et al., 2021b), generative adversarial networks (Bian et al., 2019; Assouel et al., 2018), autoregressive models (Shi et al., 2020; Popova et al., 2019; Flam-Shepherd et al., 2021). In the most recent developments, diffusion models have gained prominence in molecule generation (Hoogeboom et al., 2022; Bao et al., 2023; Huang et al., 2023; Xu et al., 2022; Wu et al., 2022), marking a novel direction in the field.

Generally, these methods can be categorized into unconditional and conditional molecule generation. Unconditional molecule generation (Hoogeboom et al., 2022; Huang et al., 2023) generates molecules without any external constraints, representing the naive form of molecule generation.

**Conditional molecule generation**, however, which conduct valid molecules that exhibit desired properties (Kang & Cho, 2019; Kotsias et al., 2020; Yang et al., 2023), is a pivotal approach of inverse molecular design (Sanchez-Lengeling & Aspuru-Guzik, 2018). Towards this end, many prior works (Hoogeboom et al., 2022; Gebauer et al., 2019; 2021) adopt the idea of conditional diffusion, having centered on learning a molecule distribution conditioned on certain properties from existing data. By sampling from this distribution with conditions aligning with desired properties, new molecules can be generated. Here we scrutinize the widely-used condition types. Many previous attempts (Hoogeboom et al., 2022; Bao et al., 2023; Huang et al., 2023) mostly employ a specific property value (*e.g.,* polarizability, dipole moment, and molecular orbital energy) as the condition in diffusion, ensuring the generated molecules adhere to the particular chemical or quantum attributes. These efforts set value-based conditions to ensure the molecules conform to certain chemical or quantum characteristics. Some studies (Gebauer et al., 2021; Kotsias et al., 2020) stipulate specific structural conditions as molecular fingerprints. However, solely specifying a target property often falls short of addressing the comprehensive demands of inverse molecular design (Honório et al., 2013; Gebauer et al., 2021; Lee & Min, 2022). To overcome this limitation, some studies (Gebauer et al., 2021; Bao et al., 2023; Yang et al., 2023) have combined that combine multiple properties as conditions. Such strategies can cater to multiple targets in inverse molecular design, such as generating molecules with low-energy structures and small HOMO-LUMO gaps. In contrast to these value-based conditional generative models confined to a single or a handful of properties, our work further proposes a text-guided method, a flexible and generalized way to control the generation process of molecules.

# B EXPERIMENT DETAILS

## B.1 EVALUATION METRICS

**Mean absolute error (MAE).** (Willmott & Matsuura, 2005) is a measure of errors between paired observations. Given the property classifier network $\phi_p$, and the set of generated molecules $\mathbb{G}$, the

MAE is defined as:

$$\text{MAE} = \frac{1}{|\mathbb{G}|} \sum_{\mathcal{G} \in \mathbb{G}} |\phi_p(\mathcal{G}) - c_{\mathcal{G}}|, \tag{16}$$

where $\mathcal{G}$ is the generated molecule, and of which $c_{\mathcal{G}}$ is the desired property.

**Novelty.** (Simonovsky & Komodakis, 2018) is the proportion of generated molecules that do not appear in the training set. Specifically, let $\mathbb{G}$ be the set of generated molecules, the novelty in our experiment is calculated as:

$$\text{Novelty} = \frac{|\mathbb{G} \cap \mathbb{D}_b|}{|\mathbb{G}|}. \tag{17}$$

**Atom stability.** (Hoogeboom et al., 2022) is the proportion of the atoms in the generated molecules that have the right valency. Specifically, the atom stability in our experiment is calculated as:

$$\text{Atom Stability} = \frac{\sum_{\mathcal{G} \in \mathbb{G}} |\mathbb{A}_{\mathcal{G}, \text{stable}}|}{\sum_{\mathcal{G} \in \mathbb{G}} |\mathbb{A}_{\mathcal{G}}|}, \tag{18}$$

where $\mathbb{A}_{\mathcal{G}}$ is the set of atoms in the generated molecule $\mathcal{G}$, and $\mathbb{A}_{\mathcal{G}, \text{stable}}$ is the set of atoms in $\mathbb{A}_{\mathcal{G}}$ that have the right valency.

**Molecule stability.** (Hoogeboom et al., 2022) is the proportion of the generated molecules where all atoms are stable. Specifically, the molecule stability in our experiment is calculated as:

$$\text{Molecule Stability} = \frac{|\mathbb{G}_{\text{stable}}|}{|\mathbb{G}|}, \tag{19}$$

where $\mathbb{G}_{\text{stable}}$ is the set of generated molecules where all atoms have the right valency.

### B.2 The Quantum Properties in QM9 Dataset

We consider 6 main quantum properties in QM9:

- $C_v$: Heat capacity at 298.15K.
- $\mu$: Dipole moment.
- $\alpha$: Polarizability, which represents the tendency of a molecule to acquire an electric dipole moment when subjected to an external electric field.
- $\varepsilon_{\text{HOMO}}$: Highest occupied molecular orbital energy.
- $\varepsilon_{\text{LUMO}}$: Lowest unoccupied molecular orbital energy.
- $\Delta_\varepsilon$: The energy gap between HOMO and LUMO.

## C LIMITATIONS

This section complements section 6 and further explains the limitations of our work as well as future directions.

As mentioned earlier, one of the main challenges we encountered during the development of TED-Mol was the scarcity of high-quality data linking real-world 3D molecules to their corresponding textual descriptions. This data deficiency has restricted our ability to fully train the model on a comprehensive set of text-3D molecule pairs, potentially limiting the model's performance in generating molecules that accurately align with complex textual descriptions.

In future work, it would be beneficial to invest in the curation of more extensive and diverse datasets. This could involve collaborative efforts with domain experts in chemistry and pharmacology to create rich, descriptive text annotations for a wide range of 3D molecular structures. Such a dataset would not only benefit TEDMol but also the broader scientific community working on text-guided molecule generation.

Another limitation of TEDMol is the relative slowness of the sampling process due to the iterative nature of the total diffusion steps. This can pose a challenge in scenarios requiring rapid molecule

generation, such as high-throughput drug discovery or material design. Future work could explore computational optimizations or alternative sampling methods to speed up the process. We believe that this limitation could be resolved in further research with more advanced generation models such as the consistency model.

Additionally, the generalization of TEDMol to more complex and real-world scenarios needs further exploration. While we have demonstrated the model's performance on the QM9 dataset and a dataset from PubChem, it would be interesting to test TEDMol on a wider variety of datasets and under more challenging conditions.

Furthermore, the practicality of our model in real-world drug design scenarios remains a subject of ongoing investigation. The current design of TEDMol necessitates that the properties to condition on must be known upfront during the training phase. This might not always be feasible in practical settings, where specific properties linked to a particular drug discovery target may only become available later on, and often with very limited sample data.

Future research could explore ways to make TEDMol more adaptable and flexible for real-world use cases. This could involve developing methods for conditioning the model on new properties post-training or improving its ability to learn from small sample sizes. These improvements could significantly enhance TEDMol's applicability and effectiveness in drug design scenarios.

We acknowledge these limitations and believe they provide valuable directions for future work. We remain optimistic about the potential of text-guided 3D molecule generation as an important tool for advancing drug discovery and related fields.

In conclusion, while TEDMol has shown promising results in text-guided 3D molecule generation, there are several areas for improvement and exploration. We believe that addressing these limitations will not only enhance the performance of TEDMol but also contribute to advancing the field of text-guided molecule generation as a whole.

