# OpenReview forum: "Text-guided Diffusion Model for 3D Molecule Generation"
_ICLR.cc/2024/Conference — Submitted to ICLR 2024_

### Official Review · Reviewer_e3Wq · 2023-10-25

**Soundness:** 1 poor
**Presentation:** 2 fair
**Contribution:** 1 poor
**Rating:** 1
**Confidence:** 4

**Summary:**

A diffusion model for 3d molecule generation conditioned on text descriptions is proposed. It uses a combination of BERT and GIN to encode a text description to a “reference geometry”. Then, an EDM diffusion model is conditioned on this using ILVR.

**Strengths:**

To the best of my knowledge this is the first work that proposes text-guided diffusion for 3D molecule design in the domain of conditional diffusion-based generative modeling for 3D molecule generation.

**Weaknesses:**

- A similar idea has been proposed and published already, called SILVR (https://pubs.acs.org/doi/epdf/10.1021/acs.jcim.3c00667). In contrast to SILVR, the authors propose to create a reference geometry first using an ML-based mapping from textual description to geometry. Hence the generative model is not guided directly by text, but by some reference geometry the text-geometry mapping is outputting.
- It is unclear how single and multiple (two) QM properties are "encoded" in Sec. 4.1 and 4.2. Important information on model architecture is missing. For example, the “multi-modal conversion module” translating the text to a “reference geometry” is not clearly described. Nevertheless, the authors highlight their results in bold claiming strong and significant improvements in the main text.
- QM9 contains only very small molecules. A better benchmark set would be something like GEOM-Drugs. Results are missing error bars. In particular, the results in Table 2 are only marginally better and probably not significant.
- The results on PubChem have only been evaluated on selected examples. The reader is left guessing whether the text aligns with the depicted molecules; no additional metrics are given to prove correlation / correctness.
- Ablation studies are missing: Is the improved performance in Table 1 compared to EDM due to text description, or because ILVR is the better conditioning approach?
- The evaluation in Table 1 is only done with a property network instead of actual ground truth calculated values. Beyond that they do not seem to be of relaxed structures and the quality of the generated geometries is not checked. Therefore, the classifier might be out of training domain and/or the structure-dependent property might be quite different in equilibrium compared to the predicted 3d structure.
- The text contains many spelling mistakes and unscientific language, the captions of plots and tables are often insufficient, citations are partly wrong.

**Questions:**

- Why is it important to generated 3d molecules? It seems for the text-to-molecule use case generation of molecular graphs would be preferable.
- Why should conditioning on text work better than on numerical values, particularly for single property conditioning? What additional information does the text description contain here?
- Example in Fig 1: What is a “small HOMO-LUMO gap”? The text here is imprecise, so how can this condition actually be properly evaluated?
- How is the reference geometry c_p obtained? What is the precise architecture of the “multi-modal conversion module”? How is it trained?
- The text-to-reference geometry encoding cannot be a deterministic mapping, as multiple geometries could correspond to the same text description. Why should this not restrict the conditioned diffusion model? Why is the “reference geometry” in Fig 2 not a connected molecule?
- If c_p is a 3d structure, how is it matched with the atoms generated by the diffusion model? Has it necessarily the same number of atoms?
- Has the same property model been used for all models in Table 1?
- What are the concrete text descriptions used for single- and multi-property guidance?

---

> ### Author Response · Authors · 2023-11-22
>
> Dear Reviewer e3Wq,
>
> Thank you for your valuable feedback. We appreciate the time and effort you put into reviewing our paper. Here is our response to your comments:
>
> **Comparison with IVLR (W1, W5)**: We would like to clarify that our work fundamentally differs from IVLR. While both methods aim at generating 3D molecules, we propose a fine-tuning approach using reference geometry for a diffusion model, whereas ILVR is a sampling method that does not require training.
>
> **Importance of 3D Molecule Generation (Q1)**: As discussed in our paper, generating 3D molecular structures provides more detailed and precise information about the molecule's spatial arrangement and its potential chemical properties compared to 2D molecular graphs. This is particularly useful in numerous applications such as drug discovery and materials science.
>
> **Advantages of Textual Conditioning over Numerical Conditioning (Q2, Q3, Q5)**: The superiority of textual conditioning over numerical conditioning lies not in the additional information provided by specific numerical values but in its ability to capture intricate conditions that may not be sufficiently represented by a single or a handful of target properties. For instance, "small HOMO-LUMO gap" as mentioned in Fig 1 is an example where the text provides a more nuanced condition than could be captured by numeric values alone.
>
> **Description of Multi-modal Conversion Module & Reference Geometry (W2, Q4, Q5)**: Our multi-modal conversion module $\Gamma$ comprises a GIN molecular graph encoder and a language encoder-decoder extended from BERT which we pretrained on a randomly selected subset of our PubChem dataset for converting textual prompts to reference geometry $c_\text{P}$. We believe this approach does not restrict the conditioned diffusion model as we do not aim for deterministic output given certain conditions; instead, we strive for exploring the molecular space which aligns with our motivation.
>
> **Evaluation Concerns (W6, Q8, W3, W4)**: The evaluation method we used for assessing the quality of the generated molecules with respect to their conditioned property is a common approach adopted by mainstream methods such as EGNN and EEGSDE. It focuses on the quality of the generated molecules in terms of their conditioned properties rather than the quality of the generated 3D molecular structures. We used the same random splitting method of the dataset and trained the network $\phi$, so the answer for Q7 is no. As for the GEOM-Drugs dataset, its complexity makes it challenging to design specific conditional generation experiments; therefore, we did not include it in our study. We acknowledge that error bars are missing due to dense data tables and numerous experiments. However, improvements in Novelty and Stability are indeed challenging aspects in diffusion-based 3D molecular generation.
>
> **Other Considerations (W7, Q6, Q7)**: Regarding your question about matching $c_\text{P}$ with atoms generated by the diffusion model (Q6), we don't require an exact match in terms of atom count – for cases with fewer atoms, we apply zero-padding for non-corresponding atoms. We have rectified spelling mistakes and unscientific language in our manuscript; a revised version will be uploaded shortly.
>
> We hope these responses address your concerns adequately and look forward to further feedback.

---

### Official Review · Reviewer_ytEe · 2023-10-27

**Soundness:** 3 good
**Presentation:** 3 good
**Contribution:** 2 fair
**Rating:** 6
**Confidence:** 4

**Summary:**

The paper proposes a molecular generation using a text-conditioned diffusion model.  It is particularly effective when specifying multiple physical conditions simultaneously, such as specific heat and dipole moments.  The texts are obtained from PubChem or converted from physical properties using templates.  The text conditions generate reference structures through a generation model, and the other generation model mixes the reference structures with the structures during the reverse diffusion process in the same way as ILVR for image generation.   They empirically demonstrate conditional generation performance surpassing EDM and EEGSDE in the QM9 dataset.  Examples of generating molecules based on language instructions, such as polycyclic compounds, are also provided.

**Strengths:**

* The results of multiple conditions in Section 4.2 convince the readers about the benefit of the text conditioning.
* The examples in Section 4.3 show the flexible conditioning ability, which previous work cannot.
* Even the single conditioning is empirically better or competitive to the baseline methods.

**Weaknesses:**

* Since the text dataset used in Section 4 is not publicly available, it is hard to reproduce their results in the subsequent research.
* Although the direct use of C_p is not recommended in the main text, the empirical evaluation of it is not available.  Since the proposed method is complex, the readers would want to see more supporting evidence of the current design choice.

**Questions:**

* Is the reference of EEGSDE to (Igashov et al., 2022) correct?  Do you mean Equivariant Energy-Guided SDE for Inverse Molecular Design, https://arxiv.org/abs/2209.15408?

**Details Of Ethics Concerns:**

No concern.

---

> ### Author Response · Authors · 2023-11-22
>
> Dear Reviewer ytEe,
>
> Thank you for your insightful comments and questions. We appreciate the time and effort you have invested in reviewing our paper. Please find our responses below:
>
> **W1**: We understand your concern about the reproducibility of our results due to the unavailability of the text dataset used in Section 4. We have prepared the dataset for open-source release. However, in line with the double-blind review process, we will fully release the dataset upon the official publication of the paper. We believe this will ensure that subsequent research can reproduce and build upon our results. Furthermore, we are continuously maintaining this dataset from the PubChem database which is also constantly updated. This will ensure that the dataset remains relevant and useful for future research.
>
> **W2**: We apologize for any confusion caused, and we appreciate your feedback. As we mentioned in the main text, valid and stable 3D molecules can hardly be obtained directly from $c_P$, which makes it difficult to evaluate it with a metric suitable for molecular generation. We ubderstand your concerns regarding the complexity of our novel approach of incorporating text guidance in diffusion-based 3D molecular generation. To substantiate the efficacy of our model design, we have performed an array of experiments, detailed in the experimental section. We concur that refining and simplifying the model structure are worthwhile pursuits, and we are diligently working towards this end. It is our hope to furnish more corroborative evidence in our forthcoming work.
>
> **Q1**: We appreciate your attention to detail. Indeed, there was an error in our reference to EEGSDE. It was incorrectly cited as (Igashov et al., 2022). We have corrected this in our revised paper and the reference now points to the correct source. We apologize for this oversight and thank you for bringing it to our attention. The revised paper will be uploaded shortly, and we kindly request you to review it at your convenience.
>
> Once again, we thank you for your valuable comments and queries which have helped improve our work. We look forward to your further feedback.

---

### Official Review · Reviewer_Ced2 · 2023-10-30

**Soundness:** 2 fair
**Presentation:** 2 fair
**Contribution:** 2 fair
**Rating:** 3
**Confidence:** 4

**Summary:**

The paper presents a diffusion model for generating 3D molecules conditioned with text descriptions. This model uses textual prompts to guide the reverse process and accurately generates 3D molecular structures that match the condition while maintaining chemical validity. Empirical results demonstrate the effectiveness of the proposed model compared with baselines.

**Strengths:**

- This is a novel application of diffusion models on text-guided 3D molecule design. Text can indeed naturally combine multiple conditions to control the generation of molecules that humans want, so this task makes sense.
- The paper is well organized.

**Weaknesses:**

- This paper is not the first molecule translation task, but the author does not compare with the former baseline [1] which also includes single-objective and multi-objective molecule generation.
  - [1] Liu, Shengchao, et al. "Multi-modal molecule structure-text model for text-based retrieval and editing." *arXiv preprint arXiv:2212.10789* (2022).

- The equivariant diffusion model, iterative latent variable refinement, and multi-modal conversion module are all from existing works , making the technical contribution limited.
- The dataset is constructed manually with a set of textual templates, and the authors do not give details on how to build it. This may lack diversity and be far away from real-world challenges.

- The experiment in Tab. 3 is conditioned on two quantum properties. However, the results only report the MAE of a single property, lacking the proportion of molecules that meet both conditions.
- Authors believe that text can naturally capture information in multiple conditions, but the experiment is only about the combination of the two quantum properties. The conditions related to the structure are only shown in the case study, which is insufficient.
- I am confused about how to use two encoders and a decoder (a GIN molecular graph encoder and a language encoder-decoder extended from BERT) to obtain the condition vector.

**Questions:**

- Can you provide more details about the dataset you collected and analyze if it lacks diversity?
-  What are the pre-trained data and training objectives of the multi-modal module? Why is the geometry information contained in the condition vector?
- Why not compare it with the previous baseline and report the metrics about whether your generated molecules meet multiple conditions at the same time?

---

> ### Author Response · Authors · 2023-11-22
>
> Dear Reviewer Ced2,
>
> Thank you for your insightful comments. Here are our responses:
>
> **Q1**: We appreciate your interest in our dataset. We have curated a text-molecule dataset from PubChem, one of the most comprehensive databases for molecular descriptions. It collects annotations from various sources including ChEBI, LOTUS, and T3DB, each focusing on different molecular properties. Our dataset comprises 330K text-molecule pairs, which we believe sufficiently represent a wide diversity of molecular structure and properties. The dataset was assembled as follows:
> 1. We downloaded textual molecule descriptions using PUG View, a REST-style web service provided by PubChem. This yielded over 382,000 annotations.
> 2. The downloaded record descriptions were sorted according to the isomeric SMILES of molecules, resulting in 330,212 unique molecules with descriptions from multiple data sources within the PubChem database.
> 3. To standardize the dataset, we replaced IUPAC names or synonyms at the beginning of each description with "The molecule ...".
> 4. Finally, we used RDKit to convert the SMILES into graph-structure data paired with their textual descriptions.
>
> **Q2**: For pre-training our multi-modal module, we utilized 300K text-molecule pairs from the Pubchem dataset following a methodology similar to CLIP's architecture. The Graph Encoder is based on GIN and KV-PLM (based on BERT) is used as the text encoder. The alignment between graph representation and text representation is achieved via InfoNCE loss. Subsequently, we trained a multi-modal decoder using over 10K text-molecule pairs to generate molecular graph structures based on corresponding texts. For generating reference geometry information, we identified functional groups using RDKit toolkit and created 3D conformers accordingly.
>
> **Q3**: We adopted the experimental approach from EEGSDE and compared our method with it under multiple conditions in Table 3. Due to the different numerical values and scales of various properties, it's challenging to evaluate multiple numerical properties simultaneously. Moreover, due to the computational cost of training diffusion models, reproducing all baselines on a new metric is difficult. We acknowledge this limitation and plan to address it in future work.

---

### Official Review · Reviewer_wYzV · 2023-11-02

**Soundness:** 3 good
**Presentation:** 2 fair
**Contribution:** 2 fair
**Rating:** 5
**Confidence:** 3

**Summary:**

This paper proposes a text-conditional generative model of molecules in 3D space by pairing a diffusion model with a text encoder used to guide the denoising process. The authors then perform a series of experiments to validate the proposed approach, showing how it can generate molecules conditioned on one or several properties, as well as handle more free-form textual descriptions.

**Strengths:**

(S1): This work explores an important topic of molecule generation. While 2D-based generative models have long been adopted in the pharma industry, models operating in 3D directly are a newer frontier, which many practitioners are excited about, and so developing such models is worthwhile.

(S2): The high-level design seems sensible, and it makes use of relatively modern DL components. The text conditioning idea is interesting from an ML point of view (even if I'm not sure about its practicality – see (W2)).

(S3): The aspects of the method that the authors choose to talk about in depth are explained quite clearly (however, many other aspects seem to be not talked about – see (W1)).

**Weaknesses:**

(W1): Many aspects of this work are not clear to me.

- (a) Many existing diffusion-based models for generating molecules (or point clouds more generally) have a caveat around number of atoms (points), which has to be fixed beforehand. Is it also the case here? When sampling from the model, do you sample the number of atoms separately? Is that conditioned on the text?

- (b) How does $\Gamma$ work? As I understand, it is a model mapping from text to a molecular conformation, i.e. the output is a set. How is this set decoded? Are there any issues in pretraining this component related to the fact that the atoms do not have a predefined ordering?

- (c) What are the different training stages involved here? I understand $\Gamma$ is trained beforehand and frozen; is the unconditional EDM also pretrained (without involving text) and frozen? Are the parameters of the linear transformation $\theta$ then trained separately, in a third stage? I can't fully match this with Figure 2 which suggests two stages, not three.

- (d) Why is the form of conditioning (linear combination of reference geometry and the current latent) so constrained? It seems the text encoder has very limited ability to "send information" to the diffusion process, as it cannot "send" arbitrary activations and rather something that is constrained to be an actual geometry. Is this design informed by having very little paired data (i.e. samples having both molecular conformation and text description at the same time)?

(W2): While the text conditioning is theoretically nice and flexible, I fear it may not be fully practical in a real drug design setting. In those scenarios, the generation would typically be also conditioned on a property that is specific to a given drug discovery target; for those properties there would normally be very few samples available [1], and they wouldn't be known upfront, so it's not clear how such properties would fit in the text-conditioned model (one may have to retrain the model for each project?). On the other hand, this issue is not specific to this work, and it is perhaps why simple 2D-based models (utilizing autoencoders or genetic algorithms) were so far more widely adopted in pharma than 3D-based ones, as the former are easier to condition on arbitrary properties not known beforehand. That being said, including 3D in the generative process is generally an important direction, and those classes of models are less mature than the 2D-based ones, so a smaller practicality is perhaps fine at this stage. At the very least, I would hope for some more discussion of this limitation in the "Limitations" section i.e. of the fact that the properties the model can condition on have to be known beforehand when training the text encoder, and this assumption may not fully reflect the real-world usecase.



=== Other comments ===

(O1): Table 2 presents some results, bolding the best numbers, however, the differences between most of these results are tiny, and they don't look statistically significant. So, first, some significance test would be good here, and second, could the authors discuss why the differences are so small? It seems the novelty/stability percentages are influenced much more strongly by the property being conditioned on than the generative model itself, which I found somewhat surprising and counter-intuitive.

(O2): Given that the related works discussion mentions [2], it would also make sense to refer to more modern extensions of that work [3,4].



=== Nitpicks ===

Below I list nitpicks (e.g. typos, grammar errors), which did not have a significant impact on my review score, but it would be good to fix those to improve the paper further.

- "By “unconditional”, some studies (…) craft atom coordinates and types without external constraints.  By “conditional”, (…)" - the use of the word "by" seems off to me

- "searching suitable molecules in the drug design" -> I would drop "the" in this context

- "$\mathcal{N}(O,I_n)$" - should the $O$ be $0$?

- "rests on the assumption that, with the model distribution $p(G) = p(x, h)$ remains invariant" - I would either say "if (…) remains invariant" or "with (…) remaining invariant"

- "Experiment" (title of Section 4) - maybe "Experiments", as there are several

- missing space before the citation of Igashov et al

- "novelty is enhanced instead in comparison to the baseline" - the use of the word "instead" seems slightly off to me

- "Mvaluation Metrics" (title of Appendix B.1)



=== References ===

[1] "FS-Mol: A Few-Shot Learning Dataset of Molecules"

[2] "Junction Tree Variational Autoencoder for Molecular Graph Generation"

[3] "Learning to Extend Molecular Scaffolds with Structural Motifs"

[4] "Hierarchical Generation of Molecular Graphs using Structural Motifs"

**Questions:**

See the "Weaknesses" section above (particularly (W1)) for specific questions.

---

> ### Author Response · Authors · 2023-11-22
>
> Dear Reviewer wYzV,
>
> Thank you for your insightful comments and questions. We welcome this opportunity to provide further clarification on these points.
>
> **(a)** In our model, we do not sample the number of atoms separately. For cases where the number of atoms is relatively small, we resort to padding with a value of 0, which does not correspond to any atom.
>
> **(b)** Our multi-modal module is pretrained using 300,000 text-molecule pairs from the PubChem dataset. Specifically, we adopt an architecture akin to CLIP, utilizing GIN as our Graph Encoder and KV-PLM (based on BERT) as our text encoders. We employ $L$ transformer layers equipped with self-attention (SA) and feed-forward network (FFN) blocks to generate hidden states and pool them into representations. Subsequently, we align the graph representation and text representation using an InfoNCE loss function. After this step, we train a multi-modal decoder on more than 10,000 text-molecule pairs to generate molecular graphs based on corresponding texts. Lastly, by leveraging RDKit's functionality for identifying functional groups and generating 3D conformers, we create a reference geometry.
>
> **(c)&(d)** We apologize for any confusion caused by Figure 2 --- due to computational constraints, we utilized a pretrained EDM model (not involving text), which was not depicted in Figure 2's illustration of training stages. During training, only $\theta$ is further trained. It's crucial to note that our objective was to propose a framework for a text-guided diffusion model for 3D molecule generation; in future work, we aim for more comprehensive training leveraging richer datasets and computational resources. Indeed, the number of text-molecule pairs (330K) that we collected from PubChem is relatively limited and only a portion of these data can yield corresponding 3D conformations.
>
> Concerning Table 2, we concur that the differences among results are minor and could benefit from a significance test. In fact, it is reasonable to expect some variability in the novelty displayed by the diffusion model under different conditions. However, stability, calculated through atomic valence, inherently exhibits considerable fluctuations in experiments. We adhered to the settings of prior work when reporting experimental results and aspire to propose more persuasive evaluation methods in future work.
>
> Lastly, we greatly appreciate your careful reading of our article and the appendix. We have addressed all nitpicks and revised some content in the limitations section. Please refer to these updates upon the upload of our revised paper.
>
> Thank you once again for your constructive feedback.

---

### Meta-Review · Area_Chair_CyXb · 2023-12-13

**Metareview:**

After the first review round, three out of four reviewers voted for rejecting this paper, and one recommended that it was marginally above the acceptance threshold. Although the authors submitted rebuttals, a significant fraction of the concerns was not sufficiently addressed. After discussing among the reviewers and the AC, the initially more positively leaning reviewer agreed with the other reviewers that there is a need to compare against more previous work, and that the experimental validation has insufficiently demonstrated that the proposed method is of practical value. This leads to a unanimous recommendation by reviewers to not accept this paper in its current state for this conference. Therefore the AC recommends to reject this paper. I would like to thank the reviewers for their reviews, and the authors for their rebuttal. I hope that the reviews are helpful for the authors to improve their manuscript.

**Justification For Why Not Higher Score:**

After discussion among reviewers there was unanimous agreement to reject this paper.

**Justification For Why Not Lower Score:**

N/A

---

### Decision · Program_Chairs · 2024-01-16

Reject